# Anatomically and Biomechanically Relevant Monolithic Total Disc Replacement Made of 3D-Printed Thermoplastic Polyurethane

**DOI:** 10.3390/polym14194160

**Published:** 2022-10-04

**Authors:** Muhammad Hanif Nadhif, Muhammad Maulana Ghiffary, Muhammad Irsyad, Nuzli Fahdia Mazfufah, Fakhira Nurhaliza, Siti Fauziyah Rahman, Ahmad Jabir Rahyussalim, Tri Kurniawati

**Affiliations:** 1Medical Physiology and Biophysics Department, Faculty of Medicine, Universitas Indonesia, Kampus UI Salemba, Jakarta 10430, Indonesia; 2Medical Technology Cluster, Indonesian Medical Education and Research Institute, Kampus UI Salemba, Jakarta 10430, Indonesia; 3Mechanical Engineering Department, Faculty of Engineering, Universitas Indonesia, Kampus UI Depok, Depok 16424, Indonesia; 4Stem Cells and Tissue Engineering Cluster, Indonesian Medical Education and Research Institute, Kampus UI Salemba, Jakarta 10430, Indonesia; 5Biomedical Engineering Program, Electrical Engineering Department, Faculty of Engineering, Universitas Indonesia, Kampus UI Depok, Depok 16424, Indonesia; 6Orthopedics and Traumatology Department, Faculty of Medicine/Ciptomangunkusumo Central Hospital, Jakarta 10430, Indonesia; 7Integrated Service Unit of Stem Cell Medical Technology, Cipto Mangunkusumo Central Hospital, Jakarta 10430, Indonesia

**Keywords:** IVD, MTDR, TPU, 3D printing

## Abstract

Various implant treatments, including total disc replacements, have been tried to treat lumbar intervertebral disc (IVD) degeneration, which is claimed to be the main contributor of lower back pain. The treatments, however, come with peripheral issues. This study proposes a novel approach that complies with the anatomical features of IVD, the so-called monolithic total disc replacement (MTDR). As the name suggests, the MTDR is a one-part device that consists of lattice and rigid structures to mimic the nucleus pulposus and annulus fibrosus, respectively. The MTDR can be made of two types of thermoplastic polyurethane (TPU 87A and TPU 95A) and fabricated using a 3D printing approach: fused filament fabrication. The MTDR design involves two configurations—the full lattice (FLC) and anatomy-based (ABC) configurations. The MTDR is evaluated in terms of its physical, mechanical, and cytotoxicity properties. The physical characterization includes the geometrical evaluations, wettability measurements, degradability tests, and swelling tests. The mechanical characterization comprises compressive tests of the materials, an analytical approach using the Voigt model of composite, and a finite element analysis. The cytotoxicity assays include the direct assay using hemocytometry and the indirect assay using a tetrazolium-based colorimetric (MTS) assay. The geometrical evaluation shows that the fabrication results are tolerable, and the two materials have good wettability and low degradation rates. The mechanical characterization shows that the ABC-MTDR has more similar mechanical properties to an IVD than the FLC-MTDR. The cytotoxicity assays prove that the materials are non-cytotoxic, allowing cells to grow on the surfaces of the materials.

## 1. Introduction

Lower back pain (LBP) is listed as one of the diseases in the Global Burden of Disease study [1]. According to Geurts et al., the annual cost to society due to LBP could reach €7911.95 per patient, which can be grouped into two cost types: healthcare (51%) and societal costs (29%) [2]. The global age-standardized prevalent number of people with lower back pain (LBP) in 2017 was 577.0 million (7.5%) [3]. LBP is the leading global cause of years lived with disability (YLDs) [3]. It was also reported that the disability-adjusted life years (DALYs) related to LBP reached 63.7 million cases [4]. One of the main contributors of LBP is the degeneration of an intervertebral disc (IVD) in the lumbar section [5,6,7].

Several implant treatments have been tried to treat lumbar IVD degeneration, including intervertebral disc replacements (IDRs) [8]. One of the forms of IDR is a total disc replacement (TDR) implant. Such an implant normally consists of metallic plates at the top and bottom area with a central part made of polymers acting as a cushion [9]. The implant is intended to be inert and provides no bioactivities to the implanted area [9]. The TDR approach has been reported to be superior compared with spinal fusions for degenerative lumbar disc disease treatments [10,11,12]. The TDR approach was proven to preserve natural motions and spinal movements [6]. However, several studies expressed skepticism due to the unclear evidence of the better clinical outcomes of TDR compared to other spinal treatments [13,14,15]. In some cases, one part of the implant was dislocated from the designated location [9]. 

Due to the mentioned issues of the TDR, studies involving tissue engineering techniques for IVD treatments have flourished, which have branched out into three different approaches. The first approach utilizes 3D (bio)printing approaches to fabricate annulus fibrosus (AF) replacements that have histological features, such as lamellar configurations [16]. The used materials have varied, from natural to synthetic polymers, which have included polycaprolactone (PCL) [17], silk fibroin [18], and polylactic fibers [19]. The second approach is more focused on the nucleus pulposus (NP) region of an IVD [20] with the idea of mimicking the native extracellular matrix (ECM) in the NP to promote regenerative responses [20,21]. The NP replacement is usually in the form of a hydrogel that is made of natural biomaterials (i.e., polysaccharides and proteins) or synthetic polymers (i.e., polyvinyl alcohol, polyurethane, and methacrylate) [22,23,24,25]. The third approach is focused on developing a tissue-engineered (TE) TDR that consists of the AF and NP regions [26,27]. The AF region is made of agarose type VII hydrogels, while the NP region is made of PCL nanofibers [26].

The tissue engineering techniques with the three mentioned approaches unfortunately face some limitations. The shape of the tissue-engineered AF replacements does not mimic the bulk anatomy of a native AF [18,27]. Some of the AF replacements are even circular in shape, significantly deviating from the shape of a real AF [18,27]. NP replacements also present some challenges. An NPR in the form of hydrogels was reported to have swelling issues, which may trigger gel fragmentation [28] and inflammation [29]. 

The TE TDR issue is similar to that of AF replacements, which is related to the bulk anatomical features [26]. The TE TDR used by Lazebnik et al. was still fabricated in a circular shape [26]. In a newer study, van Uden et al. developed a TE TDR made of PCL based on the microcomputed tomography (µCT) imaging of an IVD [30]. Although the bulk construction of the TDR was similar to a native IVD, the NP region was not clearly represented by the designed porous structure [30]. However, the comprehensive anatomical reconstitution of an IVD is necessary to develop a fully functional TDR [31,32]. 

Our group proposed a monolithic total disc replacement (MTDR) made of thermoplastic polyurethane (TPU) that was based on the anatomical and biomechanical features of IVDs. The MTDR was reconstructed from the magnetic resonance imaging (MRI) results of IVDs in the lumbar segment and subsequently featured a lattice structure [33]. The lattice-structured MTDR was fabricated using a 3D printing approach in the form of fuse filament fabrication (FFF) [33], an emerging technology for developing personalized orthopedic implants and scaffolds [34,35,36], thereby complying with the anatomical features of the patients [34,35]. PU was chosen because the material has been widely used for spine treatments including TDRs [37] due to the biocompatibility, load-bearing ability, and cushion capability [38]. The material has also been used in the previous TE IVD replacements [23,25]. 

Unfortunately, the use of PU in TDRs was only used as the middle bearing part of device. Meanwhile, the PU in the TE IVD was only developed as NP scaffolds. This is such a waste of potential for PU, considering that PU filaments were proven to be biocompatible and could be fabricated into various shapes using an FFF machine [39,40,41]. Continuing from our previous study [33], this study aims to evaluate the physical, mechanical, and cytotoxicity properties of the MTDR in accordance with the anatomy and biomechanics of native IVDs.

## 2. Materials and Methods

### 2.1. Materials

Two types of TPU 3D printing filaments from Shenzhen eSun Industrial Co., Ltd. (Shenzhen, China) were used in this study. The first material was TPU 87A, which is pure thermoplastic polyurethane. Meanwhile, the second material was TPU 95A, which consists of adipic acid (50%), propyl isocyanate (30%), and 2–3-butanediol (20%). The A code on the material name reveals the shore hardness of the material, in which a higher number indicates a higher hardness. The technical datasheets mention that the tensile strengths for TPU 87A and TPU 95A are 52 MPa and 35 MPa, respectively. Meanwhile, the elongations at the break of the two materials are 500% (TPU 87A) and ≥800% (TPU 95A). The manufacturer also provides the testing results online [42,43] according to the Restriction of Hazardous Substances (RoHS) Directive 2015/863 and the Registration, Evaluation, Authorization, and Restriction of Chemicals (REACH) Regulation No. 1907/2006 using 219 substances of very high concern (SVHC). The results show that TPU 87A and TPU 95A passed the tests. 

### 2.2. Design and Preparation

The IVDs of interest were the ones in the lumbar section (L1L2, L2L3, L3L4, and L4L5). The IVD models were segmented from an MRI-based lumbar vertebrae model that was available on the Embodi3D^®^ website (near Seattle, WA, USA) [44]. The segmentation was performed using open-source software, namely the 3D Slicer image computing platform. The detailed segmentation procedures were elaborated in our previous study [33]. The IVD segmentation results were exported in a standard tessellation (STL) file format. The segmented IVDs are presented in Figure 1.

### 2.3. Lattice Configuration

The lattice structure comprised a pattern of struts with in-between gaps making a webbing-like mesh, referring to the design used by Christiani et al. [17] that was also used in our previous MTDR study [33]. The lattice structure was designed using an Autodesk Inventor 2021 software (Autodesk, Inc., San Rafael, CA, USA) and distinguished into two configurations: a full lattice configuration (FLC) and anatomy-based configuration (ABC). As the name indicates, the FLC converted the IVD models into an all-lattice structure of IVDs. On the other hand, the ABC only converted the NP region of the IVDs into an all-lattice structure, following the anatomy of the native IVD. The AP region was kept as a rigid body. Each IVD had a different anterior-to-posterior length ratio in the AF and NP regions according to a study by Zhong et al. [45]. FLC-MTDRs and ABC-MTDRs were fabricated using the two materials (presented in Figure 2a,b, respectively). Figure 2c highlights the inner structure of the MTDR. 

### 2.4. Fabrication

The fabrication processes consisted of configuring the 3D printing parameters and the 3D printing. The printing parameters referred to our initial MTDR study [33] using a Simplify3D computer-aided design manufacturing (CAM) software [46]. The 3D printing was performed using an FFF machine and CP-01 3D printer (Creality 3D Technology Co., Ltd., Shenzhen, China) with the exact hardware setups used in our previous study [33]. 

### 2.5. Geometrical Evaluation

The geometrical parameters of each lattice configuration were evaluated so that the consistency of the printing results could be determined [46]. The evaluation used a Dinolite Edge AF4915 Series microscope (AnMo Electronics Corporation, Hsinchu, Taiwan) with 50× magnification. Before the geometrical evaluation was performed, the digital microscope was calibrated according to the guidance from the manufacturer. The geometrical evaluation measured the strut and hole dimensions. For the struts, the thickness became the point of interest. The holes were in the form of trapeziums. Therefore, the dimensions of interest included the length and width of the trapeziums, the so-called hole length, and the hole width, respectively. The three parameters were measured on the top and bottom surfaces of each material and each IVD segment. For each measured parameter and IVD segment, five sections of the MTDR were chosen randomly. Furthermore, the measured parameters were presented in the form of averages with standard deviations. The fabrication errors, i.e., the discrepancies between the measurement results and the design in the CAD software, were also calculated.

### 2.6. Wettability

Wettability or hydrophilicity is one of the indications of the likeliness of a biomaterial to accommodate cell adherence, cell growth, and cell proliferation [47,48,49]. One of the common approaches to measuring hydrophilicity is by using the water contact angle (WCA) method. The WCA method followed the ASTM D7334 standard, which is carried out by measuring the contact angle of a water droplet, the so-called sessile drop technique, on top of the surface of the MTDR [50,51,52]. The WCA measurements involved a Dinolite Edge AF4915 Series microscope (AnMo Electronics Corporation, Hsinchu, Taiwan). In this study, 5 μL of water droplets was poured on top of the TPU 87A and TPU 95A surfaces and immediately photographed after 10 s [53]. The captured images were subsequently measured using a digital arc protractor embedded in the Dinolite software. In performing the WCA measurements, three specimens were prepared for each material.

### 2.7. Swelling Test

A swelling test is of importance to determine the volumetric change of an implant due to the effect of fluidic immersion during the implantation [54]. The swelling percentage indicates the ability of an implant to maintain nutritional stability, support cell growth, and enable the selection of the optimal core content [23]. The swelling test of the MTDRs was performed in an immersive condition, as written in the ASTM D570 standard. However, instead of immersing the materials in the phosphate-buffered saline (PBS) solution, the immersion test involved the simulated body fluid (SBF). Three MTDR specimens of each material were placed in 10 mL of SBF for intervals of 1 day, 3 days, and 7 days after the immersion. The immersed MTDRs were incubated at room temperature. After the incubation, MTDRs were dried and filtered using filter paper [55]. The swelling ratio of the MTDRs was calculated using Equation (1) [25,56].
(1)$$\mathrm{SR}\;\left(\%\right)=\left(\frac{m_{s}-m_{d}}{m_{d}}\right)$$
where SR is the swelling ratio, *m_s_*→is the MTDR mass after swelling after a certain period, and*m_d_*→is the initial MTDR mass.

### 2.8. Degradation Test

The degradation test involved properly buffered 10 mL SBF (pH 7.4). The degradation test referred to ASTM F1635, which was carried out for 8 weeks and evaluated weekly. The SBF solution was designed to be similar to the body’s environment so it can project the rate of material degradation accurately. After every week of immersion in the SBF, the MTDRs were incubated at room temperature for 1 h. After the incubation, the MTDRs were dried in a drying oven at 50 °C for 48 h. These procedures were repeated until the total immersion time reached 8 weeks. The degradation rate was calculated using Equation (2) [57]:(2)$$\mathrm{DR}\;\left(\%\right)=\left(\frac{\left(m_{0}-m_{1}\right)}{m_{0}}\right)$$
where DR is the degradation rate, *m*_0_→is the MTDR initial mass, and*m*_1_→is the MTDR mass after vacuum drying for each week interval.

However, the degradation rate was shown as the remaining weight, which was calculated using Equation (3):(3)Remaining weight (%)=100%−degradation rate

### 2.9. Compressive Test

The compressive test was performed to determine the mechanical properties of TPU 87A and TPU 95A in forms of a lattice structure and full rigid body. The compressive test followed the ASTM D695 standard and utilized a Shimadzu Autograph AG-IS Universal Testing Machine (Shimadzu Corp., Kyoto, Japan) with a load cell of 50 kN. According to the standard, the materials were designed to be cylindrical with a diameter of 12.7 mm and a height of 25.4 mm. Three cylindrical specimens were prepared for each material and each form.

### 2.10. Analytical Approach

The results of the compressive tests for both types of geometry were used as parameters for the analytical approach. The resulted elastic moduli of the rigid cylinders and lattice cylinders were used to determine the compressive moduli of the composites in the ABC. The compressive modulus for each ICD section was calculated using the composite mixture rule equation from Voight. The Voight composite model determines the elastic modulus longitudinal to the material orientation. The volumes of the nucleus pulposus and annulus fibrosus were obtained using the Autodesk Inventor CAD software. The composite mixture rule is shown in Equation (4):(4)$$E_{c}=E_{l}V_{np}+E_{r}V_{af}$$
where *E_c_* is the elastic modulus of the composite, *E_l_*→is the elastic modulus of the lattice model, *E_r_*→is the elastic modulus of the rigid model, *V_np_*→is the volume of the nucleus pulposus, and*V_af_*→is the volume of the annulus fibrosus.

### 2.11. Finite Element Analysis (FEA)

The FEA was performed in the ANSYS 2021 R1 Student Version software (Canonsburg, PA, USA). By performing the FEA, the effect of the stress applied on the MTDR could be determined in the forms of the maximum von Mises stress (vMS) and maximum deformation in the MTDRs with FLC and ABC. The two parameters are of importance to determine whether the applied stress to a material is still in the tolerable range of stress and strain. When performing the FEA, the static structural mechanics module was used, following our previous FEA study of a lattice structure of various polymers, including polyurethane [58]. The structural mechanics settings were set according to the factory settings from ANSYS. A compressive load with a magnitude of 300 N was applied perpendicularly to the top surface of the MTDRs according to Kuo et al. [59]. Meanwhile, the bottom surfaces of the MTDRs were treated as fixed supports. The FEA mechanisms are illustrated in Figure 3.

### 2.12. Cytotoxicity Assay

The cytotoxicity assay was cleared from ethical issues by the Ethics Committee, Faculty of Medicine, Universitas Indonesia (Protocol Number: 22-01-0084; date of approval: 24 January 2022). The assays were in the form of direct and indirect approaches (Figure 4). Both the direct and indirect cytotoxicity assays used human umbilical cord-derived mesenchymal stem cells (hUC-MSCs) from the Stem Cells and Tissue Engineering Research Center (SCTE-RC), IMERI, Faculty of Medicine, Universitas Indonesia. The umbilical cord samples were taken from babies registered at the Department of Obstetrics and Gynecology, Cipto Mangunkusumo Hospital, after the mothers signed the informed consent [60]. The UC-MSCs was isolated using modification of multiple harvest explant method [61]. The microscopic observations during the direct and indirect cytotoxicity assays were also performed at a 40× magnification to visually confirm the cell viability on the MTDR surfaces.

Before the two assays were performed, all MTDRs were sterilized in an autoclave for 2 h. Subsequently, the complete medium for MSCs was prepared. The complete medium included penicillin–streptomycin 1% (final concentration 100U/mL) (GIBCO, 15140122), amphotericin-B 1% (final concentration 2500 ng/mL) (GIBCO, 15290026), glutaMAX 1% (GIBCO, 15290026), heparin 1%, platelet concentrate 10% (Indonesian Red Cross), and the alpha modification of Eagle’s medium (α-MEM; basal cell culture medium) (GIBCO, 51200038). All the GIBCO products were produced by ThermoFisher Scientific (Waltham, MA, USA).

In the direct assay, the MSCs at passages 3–5 were seeded in 6-well plates at a density of 100,000 cells for each well using the complete medium. The experimental cell density exceeded the minimum cell density, considering the approximate growth area in a 6-well plate (9.5 cm^2^) multiplied by the population doubling of UC-MSCs (5000 cells/cm^2^). For each material, all experiments were triplicated. Afterwards, the materials were immersed in the well plate filled with MSCs and the complete medium, which were subsequently incubated at 37 °C (5% CO_2_). In each day within the 4-day interval, the MTDRs were observed under an inverted microscope and the cells were harvested using TrypLE Select enzymes (GIBCO, 12563-011) from ThermoFisher Scientific (Waltham, MA, USA). The cells were counted according to counting chambers in a hemacytometer [62,63]. The resulted data were in the form of viability rates, which were the percentages of the viable cells per total cultured cells.

The indirect assay followed the ISO 10993-5 standard for in vitro toxicity tests [64]. This assay was a CellTiter 96^®^ AQueous Non-Radioactive Cell Proliferation Assay (Promega, Madison, WI, USA) that contained 3-(4,5-dimethylthiazol-2-yl)-5-(3-carboxymethoxyphenyl)-2-(4-sulfophenyl)-2H-tetrazolium (MTS), the so-called MTS assay [65,66]. The MTS assay started with the immersion of the MTDRs to the complete medium for 24 h, 48 h, and 72 h. Subsequently, the MSCs were seeded in a 96-well plate at a density of 10,000 cells for each well using the complete medium. The well plate was incubated at 37 °C (5% CO_2_) for 24 h. On the next day, the complete medium was changed with the media that had been immersed with MTDR specimens after 24 h, 48 h, and 72 h. Each MTDR-immersed medium was added to the well plate (in triplicate) and incubated at 37 °C (5% CO_2_) for 24 h. After the incubation, the MTS reagent was added to each well. The viability of MSCs on the surfaces of MTDR specimens was determined from the binding of the reagent with the cells, which was indicated by the light absorbance (optical diffraction) in the cells. In this study, for the MTS assay we utilized a spectrophotometer with a wavelength of 490 nm. The resulting data were in the form of viability rates, which were the percentages of the absorbance of the MTDR per the absorbance of the cell control.

The microscopic observation used an ECLIPSE Ti Series inverted microscope (Nikon, Tokyo, Japan). The MTDR specimens were observed under the inverted microscope for each interval in the direct and indirect assays. The observations during the direct assays were conducted before the cell counting, while the observations during the indirect assays were performed after the incubation of the MTDR specimens.

### 2.13. Statistical Analysis

The results are displayed as the means ± standard deviation. Student’s *t*-test was used to evaluate the results for TPU 87A and TPU 95A for the different analyses, including (1) geometrical evaluation (hole width, hole height, and strut thickness), (2) wettability, (3) welling ratio, (4) degradation rate, (5) compressive modulus (lattice and rigid structures), and (6) cytotoxicity analyses (direct and indirect results). The analysis of variance (ANOVA) test was used to compare the different experimental results for each material from the different time intervals, including the (1) swelling ratios, (2) degradation rates, and (3) direct and indirect cell viability rates. All statistical analyses were performed in Microsoft Excel 365 (Microsoft, Redmond, WA, USA).

## 3. Results

### 3.1. Fabrication

The two lattice configurations of MTDRs were successfully fabricated using a 3D printing approach. The fabrication was also feasible for the two materials. Figure 5 shows the representative MTDR fabrication results for TPU 87A for the L1L2 segment. Figure 5a shows the full lattice model of the MTDR, while Figure 5b presents the ABC-MTDR, showing parts of the nucleus pulposus at the center and annulus fibrosus at the periphery, thereby resembling the anatomical features of an IVD.

### 3.2. Geometrical Evaluation

Three geometrical features of the MTDR were evaluated for both TPU 87A (Figure 6a) and TPU 95A (Figure 6b). The hole widths at the top and bottom surfaces of TPU 87A and TPU 95A (Figure 6c) were 1.34 ± 0.12 mm and 1.31 ± 0.11 mm; and 1.54 ± 0.14 mm and 1.41 ± 0.14 mm, respectively. The average fabrication error of the hole widths in all materials and surfaces was 8.22%, which was similar to the average fabrication error of the hole heights (8.35%). The hole heights for TPU 87A and TPU 95A at the top and bottom surface (Figure 6d) were 1.15 ± 0.15 mm and 1.20 ± 0.10 mm; and 1.24 ± 0.09 and 1.17 ± 0.07 mm, respectively. The strut thicknesses in the same order (Figure 6e) were 0.81 ± 0.09 mm, 0.80 ± 0.06 mm, 0.83 ± 0.07 mm, and 0.83 ± 0.07 mm, respectively. The average fabrication error of the strut thicknesses, which was 17.03%, was notably higher than for the hole widths and heights. In terms of the materials, the average fabrication errors for TPU 87A and TPU 95A were slightly similar, at 12.21% and 10.19%, respectively. The average fabrication error for all geometrical features of the two materials was 11.20%.

The Student’s *t*-test showed that the measurement results between the top and bottom surfaces of TPU 87A for all parameters were significantly similar (*p* > 0.05). On the other hand, statistical similarity between the measurements of the top and bottom surfaces for TPU 95A only appeared on the strut thickness. Meanwhile, the measurement results for the hole widths and heights between the top and bottom surfaces were significantly different (*p* < 0.05). In terms of the comparison between TPU 87A and 95A, it turned out that the hole widths between TPU 87A and TPU 95A were significantly different (*p* < 0.05), while the hole heights and strut thicknesses were significantly similar (*p* > 0.05).

### 3.3. Wettability 

The WCA of TPU 87A was 59.69° ± 1.55°, while the WCA of TPU 95A was 47.18° ± 0.46° (Figure 7). The WCA results of the two materials were statistically different (*p* < 0.05), meaning that TPU 95A was more hydrophilic than TPU 87A.

### 3.4. Swelling Ratio

The TPU 87A and TPU 95A specimens experienced swelling from day 1, at rates of 23.38 ± 1.36% and 33.26 ± 2.34%, respectively. However, the swelling appeared to be saturated, as indicated by the similar swelling ratios for each material on day 1, day 3, and day 7 (Figure 8a), which were also proven by the ANOVA test (*p* values > 0.05). The swelling ratio for TPU 87A was significantly higher than for TPU 95A (*p* value < 0.05). The results were also in accordance with the WCA results, which meant that TPU 95A had a lower WCA than TPU 87A.

### 3.5. Degradation Rate

The TPU 87A and TPU 95A specimens degraded over time after 8 weeks of immersion in the SBF (Figure 8b). The ANOVA results stated that all degradation rates of TPU 87A and TPU 95A were significantly different within 8 weeks (*p* values < 0.05). The *t*-test, nonetheless, showed that the degradation rates of TPU 87A and TPU 95A were statistically similar in the 1st and 2nd weeks (*p* > 0.05). The statistical difference (based on *t*-tests; *p* values < 0.05) started to appear in the 3rd and continued until the final week, which showed that TPU 95A degraded more significantly than TPU 87A. Compared to the 1st week, the *t*-test showed that the weights of the two materials in the final week were significantly lower (*p* values < 0.05). In the final week, the remaining weights of TPU 87A and TPU 95A were 97.50 ± 0.19% and 96.52 ± 0.35%, respectively. Using a linear regression approach, the degradation rate formulas for TPU 87A (DR = −0.36W + 100.43) and TPU 95A (DR = −0.50W + 100.6) were obtained (DR: degradation rate; W: week). Using the formulas, TPU 87A and TPU 95A were estimated to completely degrade after 278 weeks (5 years and 4 months) and 202 weeks (3 years and 11 months), respectively. The degradation rate results also confirmed the WCA and swelling ratio results, which indicated that TPU 95A was more hydrophilic than TPU 87A.

### 3.6. Mechanical Properties

The results of the compressive tests were stress–strain curves for the lattice and rigid constructs of TPU 87A and TPU 95A (Figure 9a). The inset of the linear region of the stress–strain curve is presented in Figure 9b. For both lattice and rigid constructs, TPU 95A presented better mechanical properties. For the rigid construct, the compressive modulus of TPU 95A (56.86 ± 3.58 MPa) was significantly higher (*p* < 0.05), being approximately two-fold that of TPU 87A (24.74 ± 0.85 MPa). For the lattice constructs, the compressive modulus of TPU 87A was 4.87 ± 0.39 MPa, while the compressive modulus of TPU 95A was 12.30 ± 1.03 MPa (*p* < 0.05). In brief, the compressive modulus of the lattice construct of the two materials was approximately one-fifth of the modulus of the rigid construct.

The yield points of the rigid model could not be determined because the compressive tests were stopped when the deformation reached 50% of the total length of the specimens. On the other hand, the yield points of the lattice model could be obtained as shown in Figure 8b. The yield stress of the TPU 87A lattice (0.53 ± 0.10 MPa) was around 43% of the yield stress of the TPU 95A lattice (1.23 ± 0.32 MPa). The yield strains, however, were quite similar. The yield strain of the TPU 87A lattice (10.64 ± 1.92%) was slightly lower than of the TPU 95A lattice (13.75 ± 3.97%). 

Using the Voight composite model calculation, the theoretical compressive moduli of ABC-MTDR for each IVD segment were determined. The average compressive moduli of ABC-MTDR were 15.71 MPa (TPU 87A) and 36.62 MPa (TPU 95A), which were approximately two-thirds and three times the compressive moduli of the rigid and lattice models, respectively. The compressive moduli of ABC-MTDR for each IVD segment are presented in Table 1.

### 3.7. FEA Results

Figure 10 shows the representative snippets of the FEA measurements performed in ANSYS. Figure 10a,b present the stress profiles of TPU 87A and TPU 95A (in order) after the force was applied. Figure 10c,d illustrate the deformation profiles of the two materials TPU 87A and TPU 95A, respectively.

The FEA generated the maximum vMS for each MTDR model and material (Figure 10e). For the FLC made of TPU 87A, only the MTDR for the L4L5 segment generated a vMS value exceeding the tensile strength of TPU 87A (52 MPa). The vMS values of other FLC-MTDRs made of TPU 87A were under the tensile strength limit. For the FLC from TPU 95A, the results were divided into half. The MTDRs for the L1L2 and L3L4 segments resulted in vMS values lower than the tensile strength of TPU 95A (35 MPa), while for the L2L3 and L4L5 segments, the results were higher. In contrast with the FLC, all ABC-MTDRs for the two materials presented good results, meaning that all vMS values were below the tensile strengths of TPU 87A and TPU 95A. Based on this computational approach, all ABC-MTDRs presented good mechanical safety for the potential stress applied on the implanted MTDR.

The maximum deformation values of FLC-MTDRs and ABC-MTDRs made of TPU 87A and TPU 95A were way lower than the elongations at the break of the two materials at 500% and ≥800%, respectively (Figure 10f). Mechanically speaking, all MTDRs with different geometrical configurations and materials could restrain the potential strain implicated by the applied force on the site of implantation. 

### 3.8. Cytotoxicity

We performed direct and indirect assays. The direct assay showed the perfect cell viability rates of TPU 87A and TPU 95A (Figure 11a). For the 3-day interval, the cell viability rates of TPU 87A and TPU 95A were the same as for the control, which was 100%. No dead cells were found at any time interval. However, the cell densities between the materials and the control were not always statistically similar. The only cases where the cell densities were similar between the materials and control were at day 2 for TPU 87A (*p* value > 0.05) and at day 3 for TPU 95A (*p* value > 0.05). Otherwise, the cell densities of the control were always significantly higher than for the materials (*p* < 0.05). Similar cell densities between the two materials were observed at day 1 and day 3 (*p* > 0.05). At day 2, the cell density of TPU 87A was significantly higher than of TPU 95A (*p* < 0.05). Despite the differences in results, the 100% cell viability rates of all materials at each time interval indicated that the materials were not cytotoxic. 

The average viability rate for TPU 87A was 92.5%, with ratios ranging from 78.6% to 97.5%. Meanwhile, the average viability rate for TPU 95A was 86.5%. The range of viability rates for the 4-day interval was from 61.9% to 100%. The control had an average viability rate of 95.3%. The range of viability rates for the 4-day interval was from 87.8% to 100%.

The indirect assay also demonstrated impressive results. The absorbance rates of TPU 87A and TPU 95A were higher than control for each day (Figure 11b). For the 1-day interval, the viability rates of TPU 87A and TPU 95A were 130 ± 11% and 124 ± 7%, respectively. For the 2-day interval, the viability rate was the same at 124 ± 19% for both TPU 87A and TPU 95A. On day 3, the viability rates were 117 ± 11% for TPU 87A and 122 ± 36% for TPU 95A. The ANOVA tests showed that the viability rates from day 1 to day 3 were significantly similar for both TPU 87A and TPU 95A. The Student’s *t*-tests comparing TPU 87A and TPU 95A for day 1 and day 3 resulted in *p* values > 0.05, meaning that the viability rates of the two materials were similar. The *t*-test, however, could not be performed for the day 2 results because the viability rates for all triplications were the same.

The microscopic observations also strengthened the direct and indirect assay results. Mature MSCs were observed on each day during the direct assay in the vicinity of TPU 87A (Figure 11c) and TPU 95A (Figure 11d). The cell growth also experienced confluence. Furthermore, the white dots also indicated that some cells were in the mitotic state. Similar to the microscopic observation during the direct assay, mature MSCs were also observed during the indirect assay on a 96-well plate containing TPU 87A (Figure 11e) and TPU 95A (Figure 11f).

## 4. Discussion

Many studies have integrated lattice structures and porosities into interspinal devices, initiating bioactive capabilities for cell growth and proliferation inside the lattice [17,67,68,69,70]. Using a similar lattice configuration to this study, a porous polycaprolactone (PCL) scaffold promoted the cell attachment, cell proliferation, and extracellular matrix expressions [17]. An enhancement of the cell proliferation due to the porous structures of an IVD replacement was also presented by other studies: using electrospun PCL [71], electrospun polylactic acid/gelatin [72], hybrid of electrospun PCL and agarose hydrogel [26], and electrospun polyurethane [24].

The geometrical evaluation results for the MTDRs made of TPU 87A and TPU 95A were in accordance with the results found by Nadhif et al. [46]. Nadhif et al. evaluated the printing accuracy of TPU using the same FFF machine. It was found that the average error rate from the design was 12.28% [46], which was pretty similar to the average fabrication error rate in this study (11.10%). The error rate was almost 5-fold the result found by Chung et al., which was 2.5% [73]. However, the fabrication error rates in the metric scale between this study and the study by Chung et al. were not significant. The average fabrication error rate in this study was 0.12 mm, while the average fabrication error rate found by Chung et al. was 0.05 mm [73]. The more precise result from Chung et al. was arguably due to the use of a more sophisticated FFF machine (The Ultimaker 3, Create Education Limited, UK) than the machine used in this study.

The hydrophilicity of biomaterials, which can be determined by WCA, is one of the important factors in the biocompatibility and antibacterial nature of implants [74]. In this study, the WCAs of TPU 87A and TPU 95A were in the same range as for the 3D-printed TPU used by Kasar et al. [75] and Kucinska-Lipka et al. [41], being hydrophilic (WCA < 90°) [52,76]. Another report, on the other hand, considered the WCAs of TPU 87A and TPU 95A as being moderately wettable [41], the surfaces of which provided better adhesion for human cells and enhanced biocompatibility [76,77,78].

According to cell counting and morphology visualization, TPU 87A and TPU 95A have been proven to be biocompatible substrates for the growth and differentiation of various cells. The MSC adhesion on the surfaces of TPU 87A and TPU 95A was proven by the microscopic observations (Figure 11c–f), while the viability was confirmed by the direct and indirect assays. The good cell proliferation on the surfaces of the materials may also have been due to the absence of hazardous chemicals and SVHC in the materials, according to the RoHS and REACH results. The cytotoxicity test results conformed to previous studies that proved the human cell adhesion and viability on the surface of polyurethane using human umbilical vein endothelial cells [79,80], human bone marrow derived mesenchymal stem cells [81], and fibroblast CCL-136 cell lines [39].

In this study, hUC-MSCs were used for the direct and indirect cytotoxicity assays. The hUC-MSCs previously presented a stable morphology that did not change during culture periods [82]. This study used the hUC-MSCs with passages 3–5, the proliferation rate of which was still retained [82]. Human UC-MSCs that were obtained by this isolation method have been injected to treat patients with bone defects [83], osteoarthritis [84], and fractures [85], confirming the previous studies about the good osteogenesis capabilities of hUC-MSCs [86]. Human UC-MSCs have also been used in bone tissue engineering. In a report by Rahyussalim et al., hUC-MSCs were combined with hydroxyapatite scaffolds in the treatment of vertebral bone defects due to spondylitis tuberculosis, which suppressed the inflammatory process and immune responses caused by the tuberculosis infection [87]. Based on the abovementioned studies, hUC-MSCs can be loaded on the MTDR before implantation, allowing for regenerative treatments of the IVD.

The swelling ratio of TPU 87A (23.38%) and TPU 95 (33.26%) until day 7 was constantly diminutive, meaning that the MTDR had water retention ability. Based on a study by Urban and Maroudas, the swelling ratios of the two materials were much lower than the swelling ratio of human lumbar discs in vitro at around 300% (unloaded) and around 125% (applied pressure of 0.65 MPa), respectively [88]. A newer study confirmed the previous results that the swelling ratios of the TPUs were also significantly lower than the average swelling of human lumbar discs (142% under 1 kPa preload) [89]. The low swelling ratio, however, may not compensate for the biological performance of the MTDR. Scaffolds with a low swelling ratio tend to maintain their structural integrity [90]. Furthermore, the water retention of the materials stabilizes the size and shape of the material during the cell culture in vitro and in vivo [91,92]. The water retention is also of importance for enhancing the cell adhesion and proliferation at the surface of the implant site [91].

TPU 87A and TPU 95A require ~5 years and ~4 years, respectively, to completely degrade, meaning that both materials are both biostable yet biodegradable [93]. TPU 95A degraded faster, presumably due to the high content of adipic acid, a substance known to synthesize adipate polyols that are commonly used to synthesize biodegradable PU [93]. Clinically, the degradation of PU as an implanted scaffold must occur at an optimal degradation rate. PU has tunable physiochemical properties through its chemical design, which makes it possible to change the degradation rate characteristics [94]. Several studies have suggested methods to slow down and accelerate the degradation rates of PU [95,96]. The addition of glycerol as a chain extender and tolylene-2-4 diisocyanate (TDI) as a post-linker can reduce the hydrolytic degradation rates of PU [97]. In contrast, hydrophilic moieties such as PEG can be inserted into the soft segments of the PU backbone to accelerate the degradation [98]. However, in the future, the optimal degradation rate for PU as an implant material needs to be studied more comprehensively, especially as an IVD replacement material.

Based on the degradation calculation, the disc regeneration must be achieved in less than 4 years. The main challenge is that the MTDR should host a compatible environment for the cells and ECM. According to Erwin and Hood, the IVD can be regenerated through the provision of cellular replacement, as well as the regeneration of the proteoglycan networks or the vertebral end plates (VEPs) [99]. PU as a cellular replacement causes the degradation of esters, amides, and carbamic acid, which can help in cell and tissue metabolism [96,100,101]. Esters of carboxyl and hydroxyl groups can support cell development up to the phase of osteogenic cell formation [102]. On the other hand, peptide bonds based on amide linkages play a role in the structure of proteins, enzymes, polypeptides, and other biological molecules [103]. Based on this, hopefully the tissue regeneration approach is successful in forming a new IVD before the TPU 87A and TPU 95A are completely degraded. 

In this study, the compressive tests were performed for two constructs of samples: rigid and lattice. In the FLC-MTDR, all IVD structures were imitated using a similar configuration, which was the lattice construct. This meant that the elastic moduli of the lattice construct made of TPU 87A and TPU 95A were directly translated as the elastic moduli of the FLC-MTDR. In the ABC-MTDR, the lattice construct represented the nucleus pulposus, while the rigid structure represented the annulus fibrosus. The compressive moduli of lattice TPU 87A and TPU 95A were around five times and twelve times higher than the compressive modulus of the non-degenerate NP reported by Johannessen and Elliott (1.01 MPa) [104]. For the AF, several biomechanical tests were performed, albeit mostly using tension and not compression. As mentioned by Vergari et al. [105], O’Connell et al. estimated the elastic modulus of lumbar AF using biaxial approaches, which was 7.33 ± 5.50 MPa [106]. This meant that the elastic moduli of the AF part in the ABC-MTDR were roughly three times (for TPU 87A) and eight times (for TPU 95A) the human AF. Using the Voight composite model, we calculated the elastic moduli of the ABC-MTDR as 15.71 MPa (TPU 87A) and 36.62 MPa (TPU 95A), which were in the range that Yang et al. reported (5.8 MPa–42.7 MPa) [107]. The elastic moduli of the FLC-MTDRs made of both materials were technically still in the range reported by Yang et al. However, the homogenous structure of the configuration was not anatomically or biomechanically relevant to the real human IVDs [105,106,107]. The ABC-MTDR was the most suitable in terms of the anatomy and biomechanics. The discussion is summarized in Table 2.

## 5. Conclusions

Several points can be concluded from this study:The ABC-MTDRs made of TPU 87A and TPU 95A were successfully fabricated using an FFF-based 3D printing approach following the anatomical features of IVDs with tolerable fabrication error rates;The lattice and rigid structures mimicked the mechanical properties of the NP and AF, respectively, as indicated by the similar elastic moduli between the fabricated structures and the respective tissues. When the NP and AF features were combined as an ABC-MTDR entity, the elastic modulus of the ABC-MTDR was also similar to an entire IVD;The WCA measurement and swelling test results confirmed the hydrophilicity and water retention ability of the two materials, like the NP;The degradation rates of the two materials were projected to be no less than 4 years, which are arguably acceptable considering the prospects of an implant to support the IVD regeneration, instead of becoming a prosthesis only;The direct and indirect cytotoxicity assay results indicated that the MTDR was non-cytotoxic, allowing for cells to grow on the surfaces of the materials.

## 6. Future Perspective

The current results of this study have contributed an initial basis towards the regenerative approach in IVD treatments. For future perspectives, several enhancements should be studied regarding the geometry, degradability, biomechanics, and regenerative medicine approaches.

In this study, the biomechanical properties of the NP were reconstituted in a lattice structure according to Christiani et al. [17]. For the next studies, it might be interesting to also characterize the biomechanical performance of auxetic structures in the NP region [67]. Besides the NP region, the geometrical configurations of the AF should also be modified to imitate the AF lamellar geometries [16]. An auxetic structure made of octagonal units [19] might be worth consideration. Another possible geometrical modification of the current MTDR could be the addition of endplates at the bottom and top surfaces of the MTDR to enhance the anatomical accuracy of the MTDR [16], which accordingly would improve the fixation of the MTDR to the vertebral bodies.

In terms of the degradability, several tests should be done in the future. Micro-computed tomography, also known as micro-CT, could be performed to reconstruct the remaining structure of the MTDR in 3D [109], while liquid chromatography–mass spectrometry could be performed to quantify and distinguish the degraded chemicals [110]. In addition, the bioactivity of the degraded chemicals could be tested by culturing the chemicals with the relevant cells and media [110]. 

The in silico biomechanical analysis in this study was limited to one loading regime due to the limitations of the software. In the future, the FEA approach could be advanced by introducing more loading regimes and motions. In the paper by Kuo et al., an FEA of IVDs in the lumbar region was conducted in a single run [59], which may produce a more comprehensive simulation approach for lumbar motions. The MTDR could also be simulated with biaxial and multiaxial motions to determine the area that is the most affected by physiological stresses [106,111]. Apart from the in silico approach, the MTDR should be tested in vitro using a bioreactor, which could provide mechanical stimulations of the material [112]. The mechanical stimulations could be applied with uniaxial, biaxial, and multiaxial motions.

To develop the MTDR for regenerative medicine purposes, compatible cells could be cultured on the surface of the MTDR before implantation [113], thereby allowing for the regenerative approaches used in IVD treatments [31]. As mentioned earlier, the presence of proteoglycans in IVDs is strongly correlated with the water-binding capacity of the NP [99]. The WCA tests showed that TPU87A and TPU95A had hydrophilic properties similar to the NP [114], which may potentially support the presence of proteoglycans during implantation. Regarding the VEP regeneration, polyurethane has been reported to provide osteoconductivity [115], potentially allowing the proper fixation of the MTDR to the vertebral bodies above and below.

## Figures and Tables

**Figure 1 polymers-14-04160-f001:**
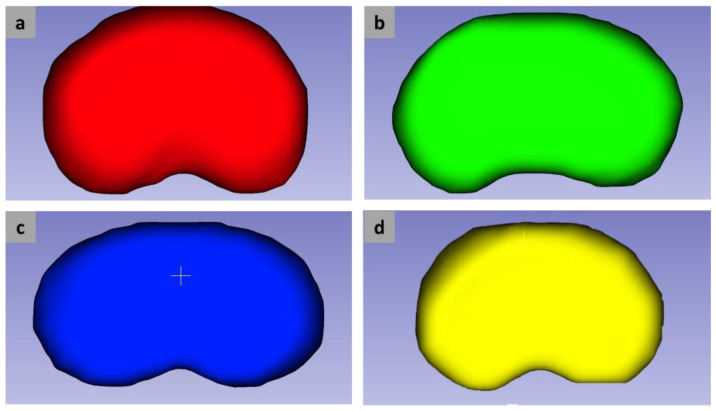
Segmentation results for IVDs in the L1L2 (**a**), L2L3 (**b**), L3L4 (**c**), and L4L5 (**d**) regions.

**Figure 2 polymers-14-04160-f002:**
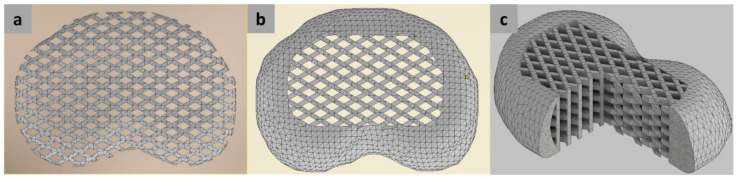
Top view of the FLC-MTDR (**a**) and ABC-MTDR (**b**). (**c**) The cross-section view of the ABC-MTDR. All 3D models were based on the reconstructed L1L2 IVD and modified using an Autodesk Inventor software.

**Figure 3 polymers-14-04160-f003:**
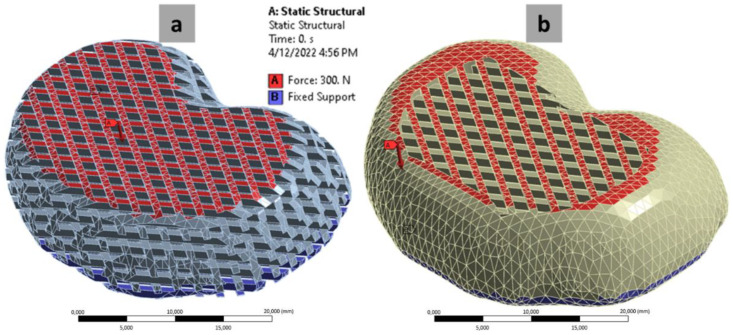
FEA mechanism illustrations of MTDR with FLC (**a**) and ABC (**b**).

**Figure 4 polymers-14-04160-f004:**
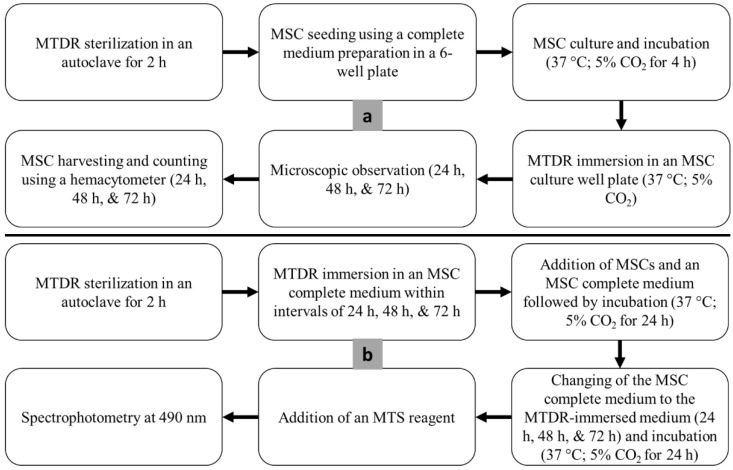
Flowcharts of the direct (**a**), and indirect (**b**) cytotoxicity assays.

**Figure 5 polymers-14-04160-f005:**
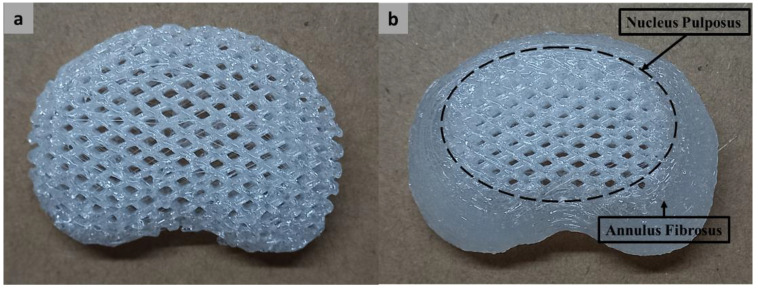
Representative products of FLC-MTDR (**a**) and ABC-MTDR (**b**) made of TPU 87A for the L1L2 segment.

**Figure 6 polymers-14-04160-f006:**
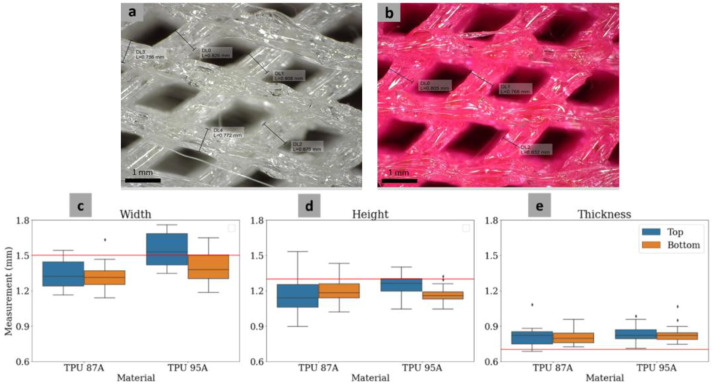
Representative photographs of the geometrical evaluation of MTDR TPU 87A (**a**) and TPU 95A (**b**). Boxplots of the measurements of the hole width (**c**), hole height (**d**), and strut thickness (**e**).

**Figure 7 polymers-14-04160-f007:**
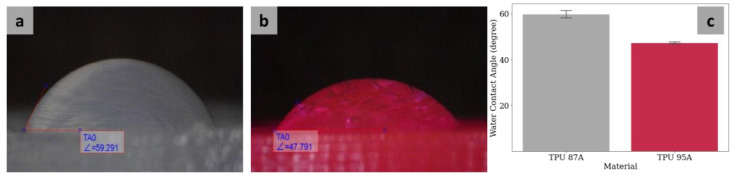
Representative photographs of TPU 87A (**a**) and TPU 95A (**b**). Water contact angle measurement results for the two materials (**c**).

**Figure 8 polymers-14-04160-f008:**
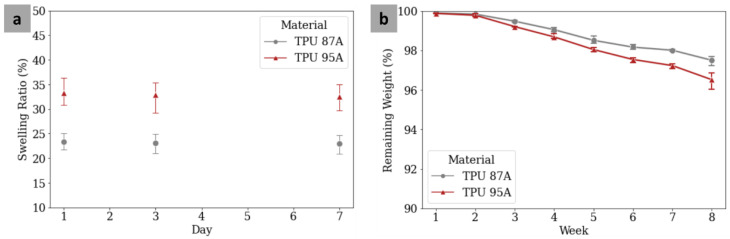
Swelling ratios of MTDR TPU 87A and TPU 95A after 1-day, 3-day, and 7-day immersions in PBS (**a**). Remaining weights of MTDR TPU 87A and TPU 95A after degradation tests using SBF for 8 weeks (**b**).

**Figure 9 polymers-14-04160-f009:**
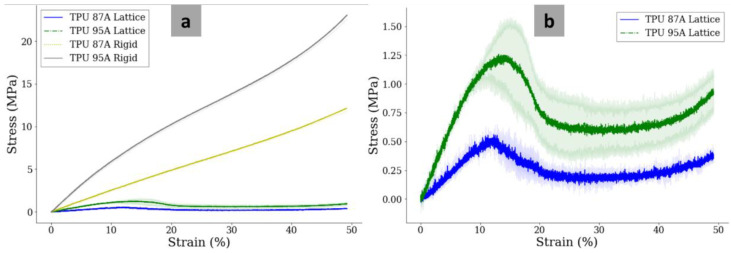
Stress–strain curves of various geometrical models of TPU 87A and TPU 95A (**a**). Inset of the stress–strain curves of the TPU 87A and TPU 95A lattices around the yield points (**b**).

**Figure 10 polymers-14-04160-f010:**
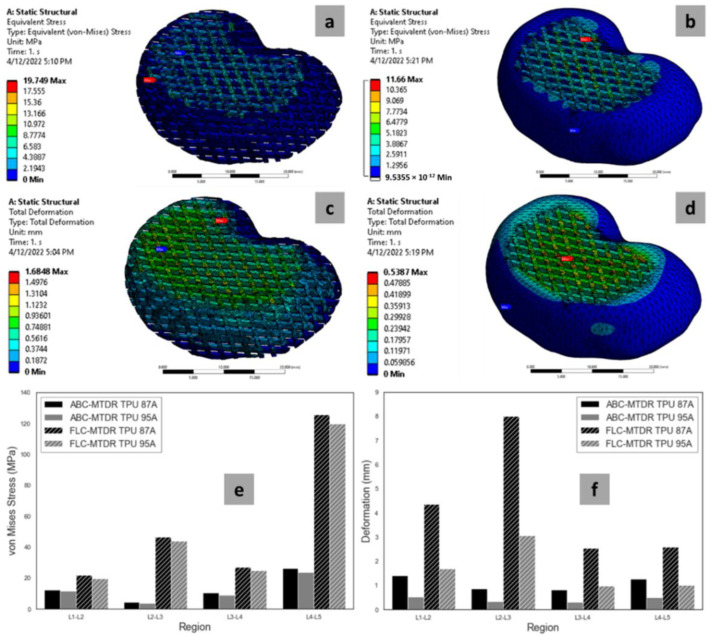
Representative stress profiles of FLC-MTDR (**a**) and ABC-MTDR (**b**). Representative deformation profiles of FLC-MTDR (**c**) and ABC-MTDR (**d**). Graphs of von Mises stresses (**e**) and deformations (**f**) resulting from the FEA.

**Figure 11 polymers-14-04160-f011:**
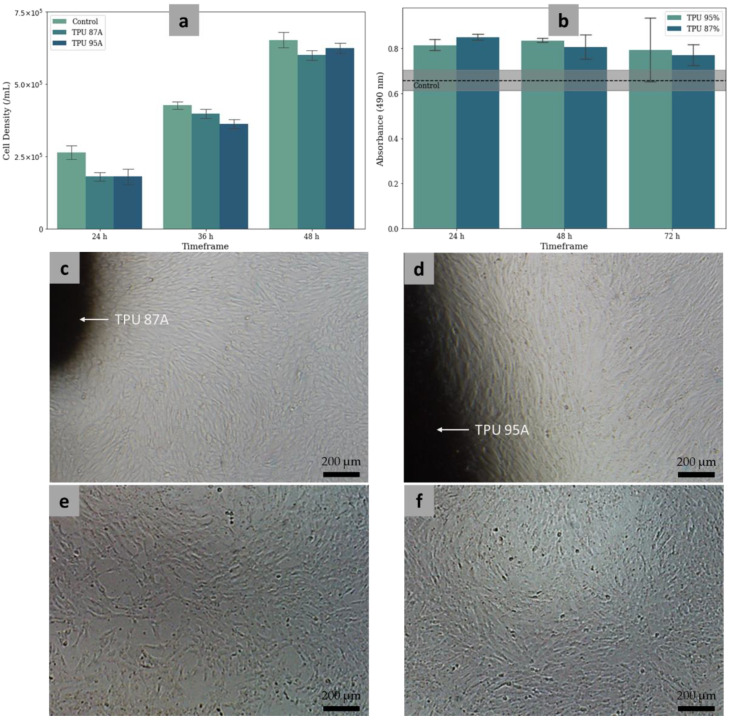
Direct (**a**) and indirect (**b**) cytotoxicity assay results. Microscopic photographs of MSCs in the vicinity of TPU 87A (**c**) and TPU 95A (**d**) in a 6-well plate after 3 days of incubation and before the direct assay. Microscopic photographs of MSCs in the 96-well plates containing TPU 87A (**e**) and TPU 95A (**f**) after 3 days of incubation and before the MTS assay.

**Table 1 polymers-14-04160-t001:** Compressive moduli of ABC-MTDR for each IVD segment based on the Voight composite model calculation.

IVD Segment	Compressive Modulus (MPa)	
	TPU 87A	TPU 95A
L1–L2	15.98	37.21
L2–L3	15.63	36.43
L3–L4	16.52	38.43
L4–L5	14.73	34.40

**Table 2 polymers-14-04160-t002:** Mechanical properties between IVD regions and our proposed research.

Mechanical Properties		Compressive Moduli (MPa)	Yield Point (MPa)	Reference
IVD	Native NP	1.01	-	[104]
		0.5–1.5	-	[108]
	Native AF	7.33 ± 5.50	-	[106]
	Native IVD (homogeneous)	20–50	-	[108]
		5.8–42.7	-	[107]
Our study	TPU 87A lattice	4.87 ± 0.39	0.53 ± 0.10	
	TPU 95A lattice	12.30 ± 1.03	1.23 ± 0.32	
	TPU 87A rigid	56.86 ± 3.58	-	
	TPU 95A rigid	24.74 ± 0.85	-	
	TPU 87A ABC-MTDR	15.71	-	
	TPU 95A ABC-MTDR	36.62	-	

## Data Availability

Available on request to the corresponding authors.

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
