# Peer review of "Anatomically and Biomechanically Relevant Monolithic Total Disc Replacement Made of 3D-Printed Thermoplastic Polyurethane"

_polymers, 2022, doi:10.3390/polym14194160_

Round 1

Reviewer 1 Report

The following comments for authors attention:

1.     LBP could reach 7,911.95 per patient. Kindly explain the readers the number specified is in dollars, euros, or rupees and so on. Similarly for Line 45.

2.     Introduction is not sufficient, authors need to clearly specify the goal with gaps in literature. Introduction section should be elaborated by conducting critical depth review.

3.     There is no specific reason given for selection of thermoplastic polyurethane rather than other biocompatible suitable for biomedical application.

4.     Any Specific reason for using shore hardness for measurement, evaluation and analysis.

5.     Figure 1 captions do not reflect in the image.

6.     Authors left out reference for the sentence “that was also used in our previous MTDR study […], The detailed segmentation procedures were elaborated by our previous study […], with exact hardware setups as our previous study […]. The printing parameters referred to our initial MTDR study […]

7.     Figure 2 explanation has not been provided in the manuscript.

8.     Is there any ASTM standards are there for swelling, degradation, wettability and if followed, please specify in the revised manuscript?

9.     No information on Figure 5 a and b explanation provided in the manuscript.

10.  It is better to provide the scale bar for Figure 6 a and b.

11.  How the p-values are calculated.

12.  How many replication experiments are conducted.

13.  Explanation of Figure 9 is not reported in the manuscript.

14.  The degradation rate of the two materials was quite slow, 4 – 5 years. However, can author add 1 comment on future work, regarding by what means authors can reduce the degradation rate.

15.  It should be better to present the conclusions in bulletin points.

Author Response

We responded to the reviewers' notes as presented in the following link.

https://docs.google.com/document/d/1g22D-cc_4Cxir9FSp2mwrp6FGaIeqwiR/edit?usp=sharing&ouid=110106128779654458686&rtpof=true&sd=true 

Reviewer 2 Report

The manuscript is interesting; the authors designed an interesting study. The results are promising, and appropriate statistical analyses were conducted as well. The authors should compare their results with the literature by the addition of at least 2 master tables. Also, i encourage the authors to add a future perspective section as well.

Author Response

We responded to your notes in the following link.

https://docs.google.com/document/d/1mmXkQ40Zk5f8y8pPan6z0_YTQQnQVQgg/edit?usp=sharing&ouid=110106128779654458686&rtpof=true&sd=true 

Round 2

Reviewer 1 Report

Congratulations for your excellent research work.

Author Response

Thank you for your resourceful and comprehensive reviews.

Here I attach the authors' responses to your reviews.

Best,

On behalf of all authors, Muhammad Hanif Nadhif
